# Hierarchical entanglement shells of multichannel Kondo clouds

Jeongmin Shim [1,2,3], Donghoon Kim[1,3] & H.-S. Sim [1] ✉

Impurities or boundaries often impose nontrivial boundary conditions on a gapless bulk, resulting in distinct boundary universality classes for a given bulk, phase transitions, and non-Fermi liquids in diverse systems. The underlying boundary states however remain largely unexplored. This is related with a fundamental issue how a Kondo cloud spatially forms to screen a magnetic impurity in a metal. Here we predict the quantum-coherent spatial and energy structure of multichannel Kondo clouds, representative boundary states involving competing non-Fermi liquids, by studying quantum entanglement between the impurity and the channels. Entanglement shells of distinct non-Fermi liquids coexist in the structure, depending on the channels. As temperature increases, the shells become suppressed one by one from the outside, and the remaining outermost shell determines the thermal phase of each channel. Detection of the entanglement shells is experimentally feasible. Our findings suggest a guide to studying other boundary states and boundary-bulk entanglement.

Boundary quantum critical phenomena[1,2] appear in gapless systems of quantum impurities[2–11], magnets with surfaces[12], edge states of topological orders[13], and qubit dissipation[14,15]. There, the presence of a boundary causes various boundary criticalities that affect the bulk, depending on boundary-bulk coupling. A character of boundaries has been revealed by the boundary or impurity entropy[16–19] that is the entropy difference between the presence and absence of the boundary. This entropy corresponds to the constant term in the dependence of the ground-state entanglement entropy on the location of the entanglement partition[18]. The entropy is a bulk quantity, as the partition is placed at long distance from the boundary, and it has been obtained by using the boundary conformal field theory (BCFT)[8–10,20–22], a standard approach for the criticalities.

While bulk quantities have been understood, boundary states are yet to be explored[23–26]. The Kondo singlet[23] in the single-channel Kondo effect, a many-body state of metallic electrons formed to screen a local impurity spin, implies that quantum entanglement between a bulk and its boundary is essential for understanding the quantum-coherent boundary-bulk coupling[27–29]. The spatial distribution of the particles forming the boundary-bulk entanglement will be a key

information of boundary quantum criticalities and related many-body effects. As the partition for the boundary-bulk entanglement is placed right at the boundary[27–30], the entanglement differs from the boundary entropy. There are difficulties in studying the entanglement. In BCFTs, the boundary degrees of freedom are absorbed into the bulk as boundary conditions, and bulk properties at long distance from the boundary are considered. Experimentally detecting entanglement typically requires inaccessible multiparticle observables. Understanding about the entanglement is desired.

Multichannel Kondo effects, where multiple channels of conduction electrons compete to screen an impurity spin, serve as a paradigm of many-body physics and boundary criticalities[6–10]. For example, in the $k$-channel Kondo ($k$CK) effect, $k$ electron channels compete to screen an impurity spin 1/2. It is described by the Hamiltonian

$$H_{k\text{CK}} = \sum_{j=1}^{k} J_j \mathbf{S}_{\text{imp}} \cdot \mathbf{S}_j(0) + \sum_{j=1}^{k} H_j. \qquad (1)$$

[1]Department of Physics, Korea Advanced Institute of Science and Technology, Daejeon 34141, Korea. [2]Present address: Arnold Sommerfeld Center for Theoretical Physics, Center for NanoScience, and Munich Center for Quantum Science and Technology, Ludwig-Maximilians-Universität München, 80333 Munich, Germany. [3]These authors contributed equally: Jeongmin Shim, Donghoon Kim. ✉e-mail: hs_sim@kaist.ac.kr

Here, the impurity spin $\mathbf{S}_{\mathrm{imp}}$ locally couples to the spin $\mathbf{S}_j(0)$ of electrons in the $j$th channel with strength $J_j > 0$, and $H_j$ describes free electrons in the $j$th channel. In the Affleck-Ludwig BCFT[8–10], the channel-isotropic case of $J_1 = \cdots = J_k$ is transformed into a free electron Hamiltonian with a nontrivial boundary condition, by mapping $H_j$ to a semi-infinite one dimension, and fusing the impurity with the boundary of the one dimension. It exhibits a boundary criticality. In channel-anisotropic cases, the competition between the channels results in quantum phase transitions[2], various non-Fermi liquids (NFLs)[6,8], and fractionalizations[31], making the effects rich. Thermal phases and their renormalization flows of the channel-anisotropic Kondo effects were experimentally observed by using quantum dots or metallic islands[32–36].

The boundary states of the Kondo effects involve a Kondo cloud[24–26] formed by the conduction electrons screening the impurity spin. Theoretically the cloud has been studied[17–19,37–39] mostly for channel-isotropic cases. For anisotropic 2CK effects, a quantity called the excess charge density was used to study a real-space structure that indicates spatial regions corresponding to the local moment and strong coupling phases[40]. However this quantity may not be suitable for quantifying the spatial distribution of a Kondo cloud, as it can be negative at certain distances from the impurity spin and even increase with the distance. The properties of the cloud, such as its channel-resolved spatial distribution, its entanglement with the impurity, its correspondence to the transition or crossover between distinct NFL phases, and its thermal suppression, are yet to be studied. It also remains unknown how to detect the clouds in the multichannel cases, while a cloud was recently observed[41,42] in the single-channel case.

The entanglement between an impurity and its Kondo cloud is a boundary-bulk entanglement[27–30]. The spatial distribution of the electrons forming this entanglement will characterize how the cloud spatially screens the impurity quantum coherently. In this work, we propose how to theoretically quantify and experimentally measure the distribution by applying a perturbation of local symmetry breaking (LSB) at a distance from the impurity. The distribution is found to exhibit channel-dependent hierarchical entanglement shells of NFL, Kondo Fermi liquid (FL), or non-Kondo FL characters in the channel-anisotropic cases. Each shell is identified by a power-law decay of the

distribution with the distance, whose exponent is determined by the scaling dimension of the boundary operator describing the character. As the temperature increases, the shells are suppressed one by one from the outside, and the remaining outmost shell determines the thermal phase of each channel. The entanglement shell structure shows that different NFLs and FLs hierarchically coexist around the boundary with spatial and energetical separation, reflecting the renormalization of the quantum-coherent impurity screening (quantified by the entanglement) in the presence of the channel competition.

## Results

### Quantifying boundary entanglement distribution

We study the entanglement negativity $\mathcal{N} \equiv \| \rho^{\mathrm{T_I}} \|_1 - 1$ between the impurity and the channels in the $k$CK effects. $\rho$ is the density matrix of the whole system, $\| \cdot \|_1$ is the trace norm, and $\mathrm{T_I}$ means the partial transpose on the impurity. This negativity is twice the conventional definition[43,44] so that its maximum value is 1. It measures quantum coherence of the screening. The screening happens by the maximal entanglement $\mathcal{N} = 1$ independent of $k$ in the channel-isotropic cases at zero temperature[30].

To quantify the spatial distribution of the entanglement, we apply an LSB perturbation breaking the Kondo SU(2) symmetry in a channel $n$ at distance $L$ from the impurity [Fig. 1a], and study the reduction $\rho_n$ of the negativity from the value $\mathcal{N}_0(T)$ in the absence of the LSB to $\mathcal{N}(L,T;n)$ in the presence of the LSB,

$$\rho_n(L,T) \equiv \mathcal{N}_0(T) - \mathcal{N}(L,T;n), \tag{2}$$

at temperature $T$. $\rho_n$ varies between 0 and 1. Larger $\rho_n$ implies that at the distance $L$ there exist more electrons participating in the entanglement. Therefore the $L$ dependence of the reduction $\rho_n(L,T)$ quantifies the spatial distribution of the Kondo cloud in the channel $n$.

The negativity has a direct relation[30] with the impurity magnetization $\mathbf{M} = \langle \mathbf{S}_{\mathrm{imp}} \rangle$ at zero temperature (Supplementary Note 1),

$$\mathcal{N} = \sqrt{1 - \frac{4\mathbf{M}^2}{\hbar^2}}, \tag{3}$$

where $\mathbf{S}_{\mathrm{imp}}$ is the impurity spin operator. This shows that the magnetization is larger as the impurity spin is less screened by, equivalently less entangled with, conduction electrons. This relation is valid at zero temperature in general situations of the Kondo effects, and it is a good approximation at low temperature $T \ll T_{\mathrm{K}}$, where $T_{\mathrm{K}}$ is the Kondo temperature.

For details, we consider a Hamiltonian $H_{k\mathrm{CK}} + H_{\mathrm{LSB}}$. The Kondo Hamiltonian $H_{k\mathrm{CK}}$ is shown in Eq. (1). Here each channel is described by free electrons in a semi-infinite one dimensional system and the impurity spin is located at the boundary of the one dimension. $H_{\mathrm{LSB}}$ describes the LSB by a local magnetic field $B$ along $x$ axis coupled to the spin $S_{n,x}(L)$ in a channel $n$ at distance $L$ from the impurity,

$$H_{\mathrm{LSB}} = BS_{n,x}(L). \tag{4}$$

In the presence of the LSB, we compute the negativity between the impurity and the channels at finite temperature by using the numerical renormalization group (NRG) method (Supplementary Notes 2-4) that we have developed[29]. We also obtain the negativity at zero temperature by using Eq. (3) and analytically computing the magnetization based on the BCFT in the presence of the LSB (Supplementary Note 5).

### Isotropic multichannel Kondo clouds

We first consider the channel-isotropic case of $J_1 = J_2 = \cdots = J_k = J$. At $T \sim T_{\mathrm{K}}$, there occurs thermal crossover from the infrared Kondo fixed point to the ultraviolet local moment (LM) phase. The Kondo phase is a

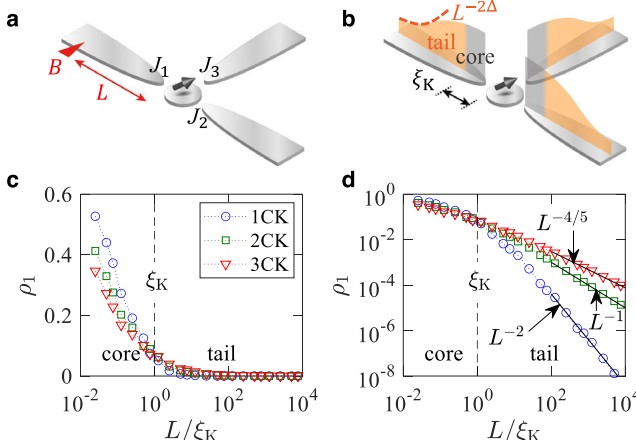

**Fig. 1 | Channel-isotropic Kondo cloud. a** An impurity spin couples to three channels with equal strengths $J_1 = J_2 = J_3$. A perturbation $B$ breaks the SU(2) spin symmetry at distance $L$ from the impurity in channel 1. The cloud distribution $\rho_1(L)$ in channel 1 is read out from the $L$ dependence of the entanglement $\mathcal{N}$ between the impurity and the channels. **b** Schematic cloud distribution. Crossover between the core and the tail happens around the cloud length $\xi_{\mathrm{K}}$. **c** Numerical renormalization group (NRG) results of $\rho_1(L)$ at zero temperature for the isotropic single-channel (1CK), two-channel (2CK), and three-channel Kondo (3CK) effects. **d** Log–log plot of **c**. The tail follows the power-law decay $L^{-2\Delta}$ in agreement with the boundary conformal field theory (BCFT).

FL in the single-channel case[4,5] and a NFL in the multichannel cases of $k \geq 2$[6,8].

In Fig. 1, the spatial distribution $\rho_n$ of the entanglement is obtained at zero temperature. The distribution extends over the whole space, having the core and the tail inside and outside the cloud length $\xi_K = \hbar v/(k_B T_K)$, where $v$ is the Fermi velocity. $\rho_n$ is much larger in the core than in the tail, showing that most electrons forming the cloud lies in the core. The core does not show any characteristics of the zero-temperature bulk criticality, strongly "binding" with the impurity. The core corresponds to the LM phase[40]. By contrast, the tail slowly decays, following the universal power law

$$\rho_n(L) \propto \left(\frac{\xi_K}{L}\right)^{2\Delta} \qquad L \gg \xi_K. \qquad (5)$$

We derive Eq. (5) using the BCFT (Supplementary Note 5), with focusing on the envelope of the Friedel oscillations in the $L$ dependence of $\rho_n$. The power-law exponent is governed by the scaling dimension $\Delta$ of the BCFT operator describing the impurity spin. For $k = 1$, $\Delta = 1$, which implies the FL of the 1CK. For $k \geq 2$, $\Delta = 2/(2 + k)$, which signifies the NFL of the $k$CK[8]. The tail accords with the bulk criticality. The exponent $\Delta$ at each phase is summarized in Table 1.

The core and tail structure of the entanglement distribution $\rho_n$ is a visualization of the quantum-coherent Kondo cloud. The LSB is useful for the visualization.

**Entanglement shells of anisotropic multichannel Kondo clouds**
We next consider channel-anisotropic cases of $k$ channels. It is known that there are multiple crossover temperatures[6]. At $T \gtrsim T_K$, the LM phase happens. At $T^* \lesssim T \lesssim T_K$, the Kondo effect by the $k$ channels ($k$CK) occurs, where $T^*$ is a crossover temperature determined by the anisotropy. Below $T^*$ there can appear $k'$-channel Kondo effects with $k' < k$. The zero temperature phase is a $k''$CK with $k'' \leq k'$ where $k''$ is the number of the channels having the largest coupling. These are shown in the phase diagrams of Figs. 2a and 3a.

**Table 1 | Scaling exponent of cloud shells**

| shell | 1CK | 2CK | 3CK | $k(\geq4)$CK | non-Kondo FL |
|-------|-----|-----|-----|--------------|--------------|
| $\Delta$ | 1 | 1/2 | 2/5 | $2/(k+2)$ | 1 |

Scaling exponent $\Delta$ of the cloud distribution $\rho_n$ of channel $n$ in the single-channel Kondo (1CK), two-channel Kondo (2CK), three-channel Kondo (3CK), $k$-channel Kondo ($k$CK; $k$ is number of channels), and non-Kondo Fermi liquid (FL) shells.

We first discuss the Kondo cloud of the anisotrpic $k$CKs at zero temperature. We find that the spatial distribution $\rho_n$ has the core and the tail of a shell structure [Figs. 2 and 3a–h]. $\rho_n$ is much larger in the core, which appears over $L \lesssim \xi_K$, than in the tail, as in the isotropic case. The tail has hierarchical multiple shells of distinct entanglement scaling behaviors. In the innermost shell, all the $k$ channels follow the power-law decay of $\rho_n(L) \propto (\xi_K/L)^{2\Delta}$ with $\Delta = 2/(2 + k)$. This shell corresponds to the NFL of the isotropic $k$CK, as identified by Eq. (5) and shown in Table 1, and appears at $\xi_K \lesssim L \lesssim \xi$ with $\xi = \hbar v/(k_B T)$. The core and the innermost shell are identical between the channels, although the coupling strengths $J_i$ are different.

On the other hand, the other shells are channel dependent. In the outermost shell, the $k''$ channels having the same coupling strength but larger than the others show different behavior from the others. These largest-coupling channels exhibit the distribution $\rho_n(L)$ of the power-law decay with $\Delta = 1$ for $k'' = 1$ (namely when one channel has stronger coupling than all the others) and $\Delta = 2/(2 + k'')$ for $k'' \geq 2$. These channels in the shell exhibit the zero-temperature $k''$CK phase, as implied by Eq. (5) (see also Table 1). The other $k - k''$ channels of weaker coupling in this shell also have nonzero distribution $\rho_n$, albeit smaller than that of the $k''$ channels. They follow the power-law decay of $\rho_n(L)$ with $\Delta = 1$, showing a non-Kondo FL that does not show the Kondo effect as discussed below. Hence the outermost shell of the Kondo cloud is composed of the NFL (resp. FL) of the $k''$CK in the $k''$ channels of the strongest coupling for $k'' \geq 2$ (resp. $k'' = 1$) and the non-Kondo FL in the other channels.

We discuss about the non-Kondo FL behavior in the $k - k''$ channels of weaker coupling. The value of $\Delta = 1$ implies that these channels are Fermi liquids. Although the value is identical to that of the 1CK case (see Table 1), these channels of weaker coupling do not exhibit Kondo behaviors. For example, in an anisotropic 2CK model[4,5,45,46], the channel of stronger coupling exhibits the $\pi$ scattering phase shift as in the 1CK case, while the weaker-coupling channel does not. It is interesting that a spin cloud, having an algebraic tail (indicated by the non-vanishing entanglement between the impurity and the channels), is developed in these weaker-coupling channels. A recent work[26] reported a similar finding that a spin cloud appears in a non-Kondo phase of a superconductor coupled with a magnetic impurity.

In Figs. 2 and 3a–h, these features of the outermost shell are shown for the 2CK of $J_1 = J + \delta J$ and $J_2 = J - \delta J$, and the 3CK of $J_{1,2} = J + (\delta J)/2$ and $J_3 = J - \delta J$. The shell appears at $L \gtrsim \xi^*$, where $\xi^* \propto |\delta J|^{-2} T_K^{-1}$ for the 2CK and $\xi^* \propto |\delta J|^{-5/2} T_K^{-1}$ for the 3CK[35]. At $L \gtrsim \xi^*$ in the 2CK, the channel 1 of stronger coupling has the 1CK FL, while the channel 2 has a non-Kondo FL. We find, using the bosonization[47,48] (Supplementary Note 6), that the channel 2 shows nonzero distribution $\rho_2$ smaller than the channel 1,

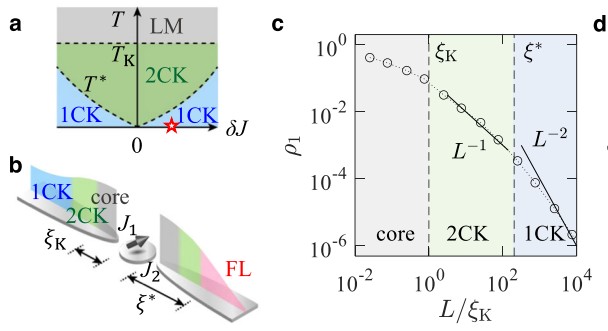
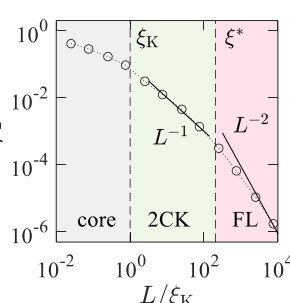
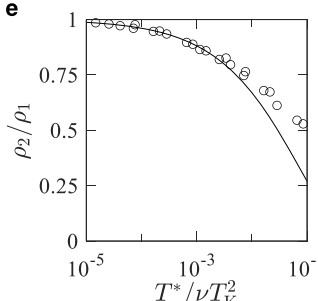

**Fig. 2 | Channel-anisotropic two-channel cloud shells. a** Two-channel Kondo (2CK) phase diagram. It consists of the local moment (LM), single-channel Kondo (1CK), and two-channel Kondo (2CK) phases. $\delta J$ is the channel anisotropy, $T$ is the temperature, $T_K$ is the Kondo temperature, and $T^*$ is the crossover temperature. **b** Cloud distribution at a point marked by the red star in the 1CK domain of the phase diagram of **a**. The coupling strengths are $J_1 = J + \delta J$ and $J_2 = J - \delta J$. The cloud distribution has the core, 1CK, 2CK and non-Kondo Fermi liquid (FL) shells. $\xi_K$ is the Kondo length and $\xi^*$ is the crossover length. **c**, **d** Log–log plots of the distribution $\rho_i(L)$ in channel $i = 1, 2$ at zero temperature. Cloud shells are identified by their power-law decay. **e** Ratio $\rho_2/\rho_1$ at $L \gg \xi^*$. Here, $v$ is the local density of states. The numerical renormalization group (NRG) results (dots) agree with the bosonization prediction (solid curve).

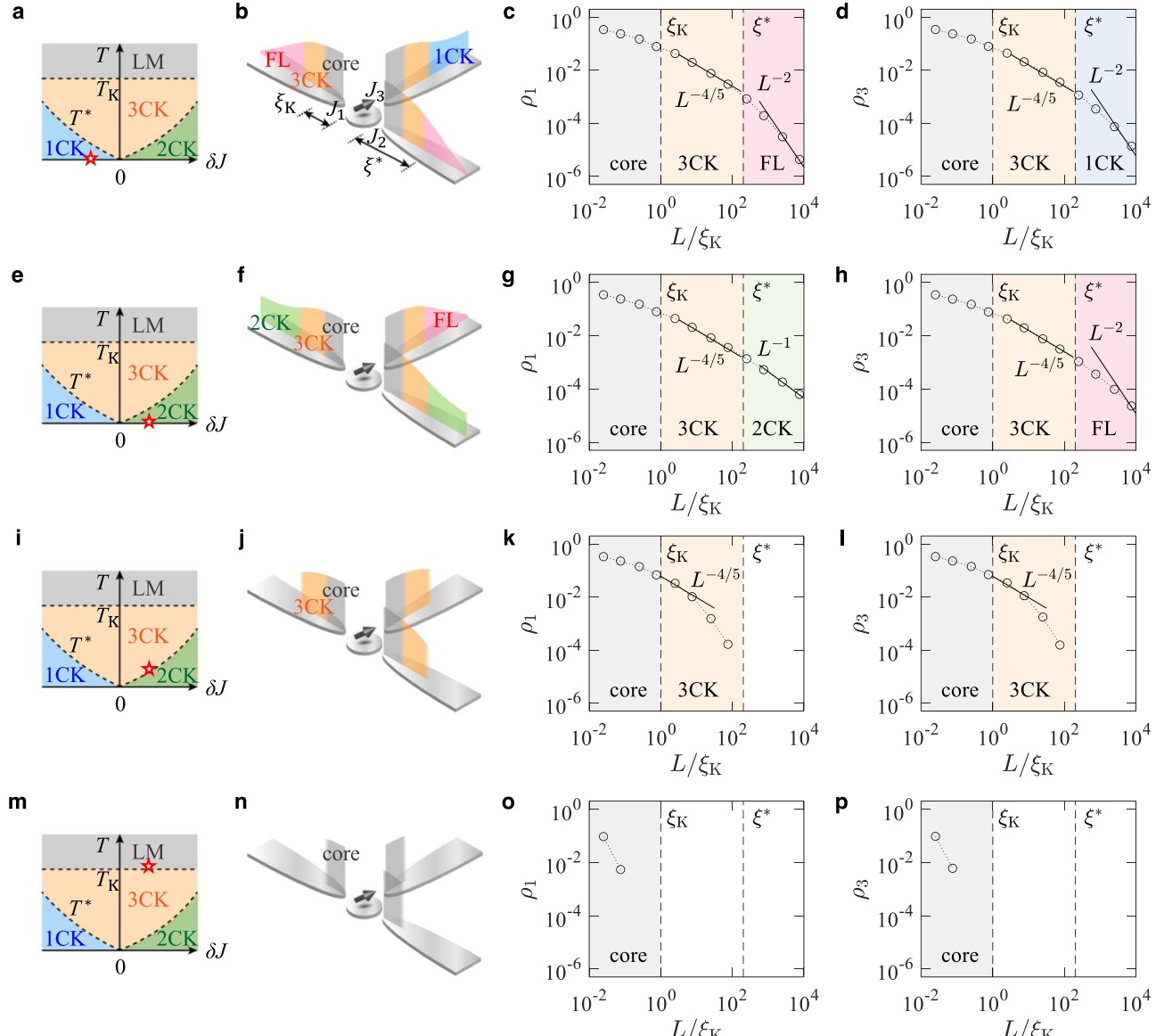

**Fig. 3 | Three-channel cloud shells and their thermal evaporation.** The three-channel Kondo (3CK) model of couplings $J_{1,2} = J + (\delta J)/2$ and $J_3 = J - \delta J$ is considered. $\delta J$ is the channel anisotropy. **a–d** The phase diagram of the model, shown in **a**, is composed of the local moment (LM), single-channel Kondo (1CK), two-channel Kondo (2CK), and three-channel Kondo (3CK) phases. At a point of $\delta J < 0$ and zero temperature $T = 0$ marked by the red star in the phase diagram **a**, the cloud distribution is drawn in **b**, the log–log plot of numerical renormalization group (NRG) results of the distribution $\rho_1(L)$ is in **c**, and the log–log plot of $\rho_3(L)$ is in **d**. $\rho_2$ is identical to $\rho_1$. In **b**, the core, 1CK, 3CK, and non-Kondo Fermi liquid (FL) shells are identified. $T_K$ is the Kondo temperature, $T'$ is the crossover temperature, $\xi_K$ is the Kondo length, and $\xi^*$ is the crossover length. **e–h** The same plots, but at a point of $\delta J > 0$ and $T = 0$. **i–l** The same plots, but at a point of $\delta J > 0$ and $T = T'$. **m–p** The same plots, but at a point of $\delta J > 0$ and $T = T_K$. As temperature increases, the outer shells disappear one by one.

following $\rho_2/\rho_1 \cong T^*/\nu T_K^2$ at $L \gg \xi^*$ [Fig. 2e]. $\nu$ is the density of states. In the 3CK with $\delta J > 0$, the channels 1 and 2 having the largest coupling exhibit the 2CK NFL in the outermost shell, while the channel 3 shows a non-Kondo FL. In the 3CK with $\delta J < 0$, the channel 3 of the largest coupling shows the 1CK FL in the outermost shell, while the other channels exhibit a non-Kondo FL.

In general anisotropic $k$CKs, there appear intermediate shells corresponding to a $q_1$CK, a $q_2$CK, $\cdots$ (from outer to inner) between the innermost and outermost shells, with the hierarchy $k'' < q_1 < q_2 < \cdots < k$ determined by the coupling strengths $J_{n=1,2,\cdots,k}$. In the shell of the $q_i$CK, the $q_i$ channels having larger coupling than the others exhibit the $q_i$CK NFL, while the other $k - q_i$ channels show a non-Kondo FL. For example, we find that in the most general case of the 3CK with $J_1 > J_2 > J_3$, the Kondo cloud is composed of the core, the innermost 3CK shell, the intermediate 2CK shell (having the 2CK NFL in two channels of larger

coupling and a non-Kondo FL in the other), and the outermost 1CK shell (having the 1CK FL in the channel of the largest coupling and a non-Kondo FL in the others) at zero temperature (Supplementary Note 4).

## Thermal evaporation of entanglement shells

To examine the thermal decoherence of the entanglement shells and hence the Kondo cloud, we compute $\rho_n(L, T)$ in Eq. (2) at finite temperatures, using the NRG. $\rho_n(L, T) = \mathcal{N}_0(T) - \mathcal{N}(L, T; n)$ quantifies the difference of the entanglement between the absence and presence of the LSB at temperature $T$; $\mathcal{N}_0(T)$ measures the entanglement that survives against thermal fluctuations at $T$, while $\mathcal{N}(L, T; n)$ measures the entanglement at $T$ further reduced by the LSB at distance $L$ in channel $n$. More reduction occurs as the impurity spin is more entangled with (i.e., more screened by) electrons at $L$. Hence, $\rho_n(L, T)$

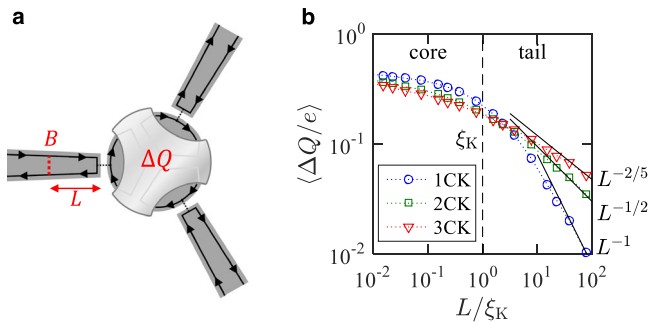

**Fig. 4 | How to detect Kondo clouds. a** A metallic dot couples with quantum Hall edge channels. Its excess charge $\Delta Q$ supports a Kondo impurity pseudospin. The cloud is formed in the channels. A local symmetry breaking (LSB) of strength $B$ is applied by placing a quantum point contact in a channel $n$ at distance $L$ from the dot. **b** Numerical renormalization group (NRG) results of $\langle \Delta Q \rangle$ as a function of $L$ for the isotropic single-channel Kondo (1CK), two-channel Kondo (2CK), and three-channel Kondo (3CK) effects. $e$ is the electron charge and $\xi_K$ is the Kondo length.

quantifies the entanglement distribution at $T$ with varying $L$. Note that in the absence of the LSB, the entanglement algebraically decays thermally[30], $\mathcal{N}_0(T) = 1 - a_k(T/T_K)^{2\Delta}$ at $T \ll T_K$, where $a_k > 0$ is a constant.

For the 3CK with $\delta J > 0$, Fig. 3e–p show the temperature dependence of the entanglement shells. Thermal fluctuations suppress shells outside the thermal length $\hbar v/(k_B T)$, while it does almost not affect shells inside. So the outer shells are thermally "evaporated" one by one. At $T \ll T^*$, the outermost shell, located at $L > \xi^*$, shows the 2CK NFL in the channels 1 and 2, as discussed above. At $T^* \lesssim T \lesssim T_K$, the outermost shell is almost suppressed. Then the remaining inner shell at $\xi_K \lesssim L \lesssim \xi^*$, whose character is the 3CK NFL, determines the thermal phase. When the temperature further increases to $T \gtrsim T_K$, only the core at $L \lesssim \xi_K$ survives and represents the LM thermal phase.

This clearly shows that the hierarchical shells of the boundary entanglement at zero temperature is the manifestation of the renormalization group flow in the development of the Kondo effects. Inner shells are "bound" more strongly with, namely more entangled with, the impurity, being more robust against thermal fluctuations. Namely, inner shells cause the boundary condition of the bulk conduction electrons of higher energies, hence, determining phases at higher temperature. Note that a related temperature dependence of a single-channel Anderson impurity model was discussed in ref. 40.

### How to detect boundary entanglement shells

Equation (3) implies that the entanglement distribution $\rho_n(L)$, hence, the Kondo cloud can be experimentally detected by monitoring the change of the impurity magnetization with varying the position $L$ of an LSB in a channel $n$. The relation is exact at zero temperature and a very good approximation at $T \ll T_K$ and $L \lesssim \hbar v/(k_B T)$ where thermal fluctuations negligibly affect $\rho_n(L)$ as demonstrated in Fig. 3.

We propose an experiment based on a charge-Kondo circuit[34,35] with which multichannel Kondo effects can be manipulated. It has a metallic dot coupled to $k$ quantum Hall edge channels (Fig. 4). Energy-degenerate charge states $|N\rangle$ and $|N+1\rangle$ of the dot form the pseudospin 1/2, and the excess charge $\Delta Q \equiv Q - (N+1/2)e$ of the dot plays the role of the magnetization $M/\hbar$ of the pseudospin. Here $N$ and $Q$ denote the number of electrons and the charge operator for the dot, respectively, and $e$ is the electron charge.

We show that a quantum point contact placed on a channel $n$ at distance $L$ from the dot results in an LSB breaking the SU(2) pseudospin symmetry (Fig. 4 and Supplementary Note 7). At $T = 0$, the negativity in the absence of the LSB is $\mathcal{N}_0(T=0) = 1$[30], while the

negativity in the presence of the LSB is $\mathcal{N}(L, T=0; n) = \sqrt{1 - 4\langle \Delta Q/e \rangle^2}$ [see Eq. (3)]. These give $\rho_n(L, T=0) = \mathcal{N}_0(T=0) - \mathcal{N}(L, T=0; n) = 1 - \sqrt{1 - 4\langle \Delta Q/e \rangle^2} \simeq 2\langle \Delta Q/e \rangle^2$ for small $\langle \Delta Q/e \rangle \ll 1$. At low temperature $T \ll T_K$ where thermal fluctuation on $\langle \Delta Q/e \rangle^2$ is negligible, $\rho_n(L, T)$ can be approximated as the zero-temperature value of $\rho_n(L, T=0) \simeq 2\langle \Delta Q/e \rangle^2$. It is possible to measure $\Delta Q(L)$, hence $\rho_n(L)$, by monitoring electric current through another quantum point contact[49] nearby the dot. The entanglement shells in isotropic and anisotropic $k$CKs can be experimentally identified with realistic parameters (Supplementary Note 7).

## Discussion

Our work demonstrates how a spin cloud screening a local magnetic impurity in a metal differs at the fundamental level from a charge cloud screening an excess charge. For the demonstration, we developed a theory of the boundary-bulk entanglement in multichannel Kondo effects. Utilizing an LSB, the spatial distribution and thermal suppression of the entanglement can be computed and experimentally detected. The distribution is a visualization of the spatial and energy structure of the quantum-coherent Kondo spin screening cloud.

The boundary-bulk entanglement is applicable to general boundary quantum critical phenomena as below. The entanglement quantifies the quantum-coherent coupling between the boundary and the bulk in boundary criticalities. Its spatial structure will have information of competing phases or boundary conditions, as suggested by the hierarchical shells of Kondo clouds. In spin-1/2 boundary criticalities, it is obtained, using the boundary magnetization and Eq. (3). In more general cases, it may be calculated with BCFT boundary operators[30].

An LSB that breaks the boundary-bulk coupling symmetry will be useful for identifying the boundary structure of boundary criticalities. The spatial structure is estimated by the change of the entanglement as a function of the location of the LSB, while the partition for the entanglement is placed at the boundary. This differs from the usual way[16] where an entanglement is studied with placing the entanglement partition in the bulk.

The boundary-bulk entanglement will be experimentally accessible. As in Eq. (3), it may have a simple relation with a boundary observable, when the entanglement has a simple form like Kondo singlets near a fixed point of boundary criticalities. Such a simple relation between an entanglement and an observable is rare. It is another usefulness of the boundary-bulk entanglement.

We anticipate that the boundary-bulk entanglement is an essential aspect of boundary criticalities and related effects such as Kondo lattices and heavy fermions[50–52].

## Data availability

All the calculation details are provided in Supplementary Information.

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

## Acknowledgements

We thank Frederic Pierre and Jan von Delft for useful discussions. This work is supported by Korea NRF via the SRC Center for Quantum Coherence in Condensed Matter (Grant No. 2016R1A5A1008184 and RS-2023-00207732) and Grant No. 2023R1A2C2003430. D.K. acknowledges support by Korea NRF via Basic Science Research Program for Ph.D. students (2022R1A6A3A13062095).

## Author contributions

J.S. performed the NRG. D.K. performed the BCFT and bosonization calculation. H.-S.S. supervised the project. All the authors were involved in developing the theoretical approach and preparing the manuscript.

## Competing interests

The authors declare no competing interests.
