## [Peer Review File · Nature Communications]

Report on manuscript NCOMMS-22-39861-T

Using the tools of numerical renormalization, bosonization, and boundary conformal field theory, the authors exploit the entanglement measure of *negativity* to uncover the spatial structure of screening clouds of the multichannel Kondo effect. When the channel symmetry is broken, shells exhibiting distinct scaling properties are found to coexist in a cloud, with the particulars depending on temperature and the number of channels k . An experiment to test the predictions by the authors is proposed, using a quantum dot coupled to k quantum Hall edge channels where the two degenerate charge states of the dot emulate the $s=1/2$ spin states of a quantum impurity.

This is a topical and very interesting paper, addressing the important issue of how to understand the spatial structure of Kondo clouds. In fact, considering the innovative and careful analysis of the authors, I think this paper may serve as a new benchmark for theoretical discussions of quantum impurity screening clouds. The paper is well written and laid out, and the results are presented in a transparent way. Provided that the points that I raise below are prudently taken into account by the authors, I will recommend publication in Nature Communications.

1) The authors' effort (in the introduction to the paper) to embed their work in the wider context of boundary-bulk entanglement problems (of which Kondo physics is but one example) is commendable. However, a drawback is that a reader not familiar with the Affleck-Ludwig BCFT approach to the multichannel Kondo effect may get lost. Given the title and the core of the paper, I think the authors owe it to the reader to add a paragraph early on where the Affleck-Ludwig approach is briefly summarized ("... reduction to 1D, folding the line to a half-line with a boundary, representing the impurity by a renormalized boundary condition...", etc., etc.), including references. In this context, I was surprised not to find the seminal paper by Andrei and Destri (Phys. Rev. Lett. 52, 364 (1984)) among the background references.

2) I think the statement "... presence of a boundary causes various critical behaviors in the bulk..." (p1) may be misunderstood. While it is true that e.g. the tail of the Kondo cloud exhibits critical scaling in the bulk, a boundary cannot *cause* bulk criticality *per se* (a singularity in the extensive term of the free energy/ground state energy, symmetry breaking, etc). While the statement refers to "critical behaviors *in* the bulk", not "critical behavior *of* the bulk" (which would be outright wrong), I would still suggest a reformulation.

3) Eq. (2) is key to the interpretation of the entanglement cloud as a screening cloud. However, I can find neither the formula nor its derivation in [31] (which is referenced to). Is "[31]" misprinted? In any event, considering the importance of Eq. (2) for the validity of the results in the paper, and also considering its nontriviality, a published proof must be available to the reader. (Actually, not having seen Eq. (2) before, I find it quite remarkable!)

4) I cannot find the proof of Eq. (4) in the Supplemental Material (as referenced to). Is it somehow hidden or implied in the BCFT discussion of the magnetization in the Supplemental material? If so, I suggest that it is lifted out of this context and be given a careful separate treatment.

5) Ref. [7] ought to be cited after the sentence "...which signifies the non-Fermi liquid of the k CK." (p3)

6) How should one understand the statement "... $\rho_n(L,T)$ quantifies the part of the entanglement distribution robust against thermal fluctuations with the help of the LSB." (p4) ? This is a cryptic statement. The only interpretation that comes to my mind is that since the LSB perturbation has already wiped out much of the entanglement, thermal fluctuations can do no worse. However, this can't be quite right! Please clarify and/or rewrite.

7) The formula for ρ_n applied to the charge-Kondo circuit is the same as the formula for $\{\text{cal N}\}$ in Eq. (2) (with M^2 replaced by $\langle \Delta Q/e \rangle^2$. How can $\{\text{cal N}\}$ be replaced by ρ_n (as defined in Eq. (1))?

8) In the treatment of the channel anisotropic 2CK effects (p14, Supplemental material), the authors use the bosonization approach pioneered by Emery and Kivelson (reference missing!). However, the refermionization in the Emery-Kivelson approach works only for a special value of the z-component of the spin exchange (analogous to the Toulouse limit of the ordinary Kondo problem). I fail to see how the authors have implemented this in their analysis.

Reviewer #2 (Remarks to the Author):

The Kondo effect with many-channels forms a screening cloud with a power-law decay at large distance. The exponent of the power law is governed by the dimension of the leading irrelevant operator acting at low energy.

The present manuscript explores this long-distance decay by evaluating the entanglement negativity - one particular marker of entanglement - between the quantum Kondo impurity and the bath electrons, with a distant local magnetic field breaking $SU(2)$.

For an anisotropic Kondo interaction, a shell structure is nicely discovered with different exponents depending on the distance to the impurity and depending also on the channel where the local magnetic field is applied. This shell structure in position and channel space mirrors energy-dependent crossovers with different characteristic temperatures occurring in the anisotropic multi-channel Kondo model.

Eventually, the way the shell structure gradually disappears with temperature is determined and an experimental setup to measure the shell structure is examined.

The paper proposes a very intuitive and simple methodology (also novel) to study impurity-bulk entanglement and the cloud structure of Kondo screening. It allows the authors to use both analytical (bosonization) and numerical (NRG) methods to solve the problem, and the approach could be readily applied to other impurity problems.

I would recommend publication of this paper but I have remarks which I believe could improve the quality of the paper.

the bosonization procedure seems to obtain, in addition to the power-law decay, Friedel oscillations in the entanglement negativity. Such oscillations are absent in the numerics, if I am not mistaken. It would be informative to state why this is the case.

The discussion on the shell of anisotropic multichannel Kondo clouds (page 3) is not easy to read and follow. I think the authors should first discuss the different possible scalings : non-Fermi liquid with a certain exponent, Fermi liquid with a Kondo screening and what they call non-Kondo Fermi liquid (it is not straightforward to understand what this last phase is). After defining these different possibilities, a table that shows which scaling applies at which distance - and in which channel - would be more easy to read than the current text.

Still on the non-Kondo Fermi liquid, it would be useful to discuss more clearly what this phase/behavior is. In particular, the authors mention an absence of π ($\pi/2$?) phase shift but, as far as I can read, they don't show explicitly this absence.

Reviewer #3 (Remarks to the Author):

The authors study theoretically the real-space entanglement structure of the multi-channel Kondo model, using a combination of numerical techniques (NRG) and analytical methods (BCFT). The entanglement negativity between the impurity and the rest of the conduction electron baths is computed. Real-space properties are deduced by comparing the negativity with and without a local perturbation applied to the bath at a distance L away from the impurity. This is done at zero temperature as well as finite temperature, in the $k=1,2,3$ channel Kondo models. Characteristic scaling behaviors are observed, which distinguish the different models. An experiment is proposed and briefly discussed, which is argued could probe the negativity scaling. The main conclusion is that for multi-channel Kondo models, there are a set of real-space 'shells' surrounding the impurity, each characterized by distinct behavior, and that this is picked up by the particular entanglement measure considered in this paper.

I do not believe that this work contains sufficient new material to be published in Nature Communications -- neither in terms of the methodology nor the physical insights drawn from them.

The main message drawn from the study is the 'hierarchical shell' structure of multichannel Kondo models in real-space. However, this is known already from Ref [41], which is cited in the introduction of the present work, but then not mentioned again despite its obvious relevance. Ref [41] does not consider entanglement, as in the present paper, and so there are certainly differences, but the notion of real-space shells around the impurity is not new.

To conclude the introduction setting out the main finding:

"This shows that different non-Fermi and Fermi liquids hierarchically coexist around the boundary with spatial and energetical separation, reflecting the renormalization of the quantum coherent impurity screening in the presence of the channel competition."

However, this is already known.

Furthermore, deeper insights are provided in Ref [41]: the shells correspond to real-space regions described by the RG fixed points, with RG flow to lower energies corresponding to real-space flow away from the impurity. In particular, this leads to the conclusion that the shell referred to in this work as the 'core' is actually the local moment shell. This explains the observation in the present work that the core behavior is very similar for multichannel models with different numbers of channels, independent of critical behavior at larger distances ("the bulk does not show any characteristics of the zero-temperature bulk criticality, strongly "binding" with the impurity"). This is because the local moment fixed point is common to all the models considered. Note also that Ref

[41] does consider the $k=2$ two-channel Kondo model and discusses the 2CK Kondo cloud, as well as the crossover in real space to Kondo Fermi liquid and non-Kondo shells due to symmetry breaking perturbations. Ref [41] also considered the "thermal evaporation" of the shells at finite- T .

The physical picture is therefore already known. This should be acknowledged and discussed in the paper. I accept that the present work provides more details on the shells in terms of their entanglement structure, as probed by the quite specific measurement setup described. This is surely interesting to specialists and deserves to be published in a more specialized journal.

In terms of methodology, again the present work does not really do something very new. The NRG technique used is sophisticated, but was developed in Ref [30] already, and used several times since to apprehend entanglement negativity properties of Kondo problems. Eq. 2 relates the $T=0$ negativity to magnetization, which is a very nice result -- but this was derived already in Ref [31]. In Ref [31], finite- T negativity between the impurity and the channels for multi-channel Kondo models was already studied. The innovation here is to study this same quantity with the same methods, but now with a local perturbation at a distance L away from the impurity. This gives the possibility to discuss negativity changes as a function of the lengthscale L . Something similar was proposed in Refs [19,20] to probe the Kondo cloud (there the focus was measurable conductance, here the authors discuss the negativity).

With $\rho_n(L,T)$ the negativity difference with and without the perturbation at site L , the authors state:

"Larger ρ_n implies that at the distance L there exist more electrons participating in the entanglement."

This seems crucial to the conclusions drawn, but is unsubstantiated. The negativity calculated is that of the impurity with the entire baths; adding the perturbation at position L can change the entanglement profile of the channels in real-space, which is then traced out. It is not clear to me that a perturbation at position L tells us about entanglement at lengthscale L . Philosophically, we can incorporate the perturbation at position L into the definition of a new non-interacting conduction electron bath, which will then have an energy-dependent hybridization function coupling to the impurity. The perturbation at L will lead (in 1d, which I guess is the setup used) to a feature at an energy scale $\omega_p \sim 1/L$ in the hybridization function. If this perturbation scale ω_p is much smaller than the Kondo scale T_K , then there will be very little effect. If $\omega_p \gg T_K$ then the Kondo effect will be affected. It seems that one can argue the results based only the energy-structure of the hybridization function, and don't need to invoke a length-scale at all! For example, one would get the same ρ_n and deduced entanglement structure in a different system that just happens to have the same energy-dependent hybridization function, where the length scale L has no meaning or analogue.

Finally, the authors propose and experiment to measure the entanglement signatures discussed in the paper. However, this potentially interesting section is not elaborated on much and is hard to follow, especially for non-experts. The main message however, is that a *charge-Kondo* setup allows to get an estimate on the negativity via dot charge changes, with and without a perturbation in the leads. This is an interesting idea, but unfortunately I do not think is practical, because the charge differences in the regime of interest are extremely small (that is because the interesting power-laws in ρ_n set in at large L , which means very small ρ_n in practice). This would be well below the charge detection noise level in an actual experiment. Furthermore, the scaling requires changing L , but a real device would have a fixed quantum point contact at a specific L , which is a physical property of the constructed device. One could make several different devices with different L , but the other inevitable differences between different devices would confound any comparison -- again especially when considering the very small numbers involved. Therefore, the statements by the authors that "suppression of the entanglement can be ... experimentally detected" and "The boundary-bulk entanglement will be experimentally accessible" are wildly overstated. This would in practice be very difficult or even impossible, and there is certainly no evidence that this is doable.

Some smaller points:

- 1) the language and presentation needs to be improved throughout
- 2) In the intro: "While bulk quantities have been understood, boundary states are yet to be explored." This is not true, they have been studied in the existing literature, as mentioned above, and in other works.
- 3) In the intro: "The properties of the cloud, such as its channel-resolved spatial distribution ... are yet to be studied." This is wrong, it has been studied before, with many of the same conclusions! See e.g. [41].
- 4) The form of the conduction electron Hamiltonians should be specified. In particular, are the authors explicitly considering 1d systems? This should be mentioned if so!
- 5) The LSB consists of a field B in the bath. What is the strength of B , and how do the results depend on B ? If B is supposed to be infinitesimal, how is this dealt with in NRG? This should be mentioned in the main text.
- 6) "In the core, ρ_n rapidly decays with L , showing that most electrons forming the cloud are in the core." I am not convinced this is implied. And furthermore, the decay of ρ_n is more rapid in the tails, as shown in Fig 1! The powerlaw involves a more negative power in the tail than core, so surely its decay is more rapid *outside*. I would say the region inside the core is the "local moment cloud".
- 7) Original references should be given to published literature where the crossovers in 3CK to either 1CK or 2CK depending on the sign of δJ are discussed. This is not a new finding.
- 8) In the conclusion, it is stated that "In spin-1/2 boundary criticalities, [the entanglement structure] is obtained using the boundary magnetization, Eq 2."

This is only true at $T=0$ and not general. The finite- T result cannot be simply related to a boundary observable.

Referee #1's overall comment: This is a topical and very interesting paper, addressing the important issue of how to understand the spatial structure of Kondo clouds. In fact, considering the innovative and careful analysis of the authors, I think this paper may serve as a new benchmark for theoretical discussions of quantum impurity screening clouds. The paper is well written and laid out, and the results are presented in a transparent way. Provided that the points that I raise below are prudently taken into account by the authors, I will recommend publication in Nature Communications.

Response: We thank Referee #1 for the positive recommendation and the constructive comments. We revise our manuscript, taking into account the comments.

Referee #1 comment-1: The authors' effort (in the introduction to the paper) to embed their work in the wider context of boundary-bulk entanglement problems (of which Kondo physics is but one example) is commendable. However, a drawback is that a reader not familiar with the Affleck-Ludwig BCFT approach to the multichannel Kondo effect may get lost. Given the title and the core of the paper, I think the authors owe it to the reader to add a paragraph early on where the Affleck-Ludwig approach is briefly summarized ("... reduction to 1D, folding the line to a half-line with a boundary, representing the impurity by a renormalized boundary condition...", etc., etc.), including references. In this context, I was surprised not to find the seminal paper by Andrei and Destri (Phys. Rev. Lett. 52, 364 (1984)) among the background references.

Response: We thank Referee 1 for the comment for general readers. We add a summary of the Affleck-Ludwig BCFT approach around Eq. (1) of the main text, and cite the paper by Andrei and Destri.

Referee #1 comment-2: I think the statement "... presence of a boundary causes various critical behaviors in the bulk..." (p1) may be misunderstood. While it is true that e.g. the tail of the Kondo cloud exhibits critical scaling in the bulk, a boundary cannot *cause* bulk criticality *per se* (a singularity in the extensive term of the free energy/ground state energy, symmetry breaking, etc). While the statement refers to "critical behaviors *in* the bulk", not "critical behavior *of* the bulk" (which would be outright wrong), I would still suggest a reformulation.

Response: To clarify the issue, we revise the sentence as "... presence of a boundary causes various boundary criticalities that affect the bulk".

Referee #1 comment-3: Eq. (2) is key to the interpretation of the entanglement cloud as a screening cloud. However, I can find neither the formula nor its derivation in [31] (which is referenced to). Is "[31]" misprinted? In any event, considering the importance of Eq. (2) for the validity of the results in the paper, and also considering its nontriviality, a published proof must be available to the reader. (Actually, not having seen Eq. (2) before, I find it quite remarkable!)

Response: Following the useful suggestion, we add the derivation in Supplementary Note 1.

Referee #1 comment-4: I cannot find the proof of Eq. (4) in the Supplemental Material (as referenced to). Is it somehow hidden or implied in the BCFT discussion of the magnetization in the Supplemental material? If so, I suggest that it is lifted out of this context and be given a careful separate treatment.

Response: We add the derivation at the end of Supplementary Note 5A and 5B.

Referee #1 comment-5: Ref. [7] ought to be cited after the sentence "...which signifies the non-Fermi liquid of the k CK." (p3)

Response: Indeed. We cite Ref. [7] after that sentence.

Referee #1 comment-6: How should one understand the statement "... $\rho_n(L,T)$ quantifies the part of the entanglement distribution robust against thermal fluctuations with the help of the LSB." (p4) ? This is a cryptic statement. The only interpretation that comes to my mind is that since the LSB perturbation has already wiped out much of the entanglement, thermal fluctuations can do no worse. However, this can't be quite right! Please clarify and/or rewrite.

Response: $\rho_n(L,T) = \mathcal{N}_0(T) - \mathcal{N}(L,T;n)$ is the difference of the entanglement between the absence and presence of the LSB at temperature T . $\mathcal{N}_0(T)$ measures the entanglement that survives against thermal fluctuations at T . $\mathcal{N}(L,T;n)$ measures the entanglement at T further reduced by the LSB at distance L in channel n . More reduction occurs as the impurity spin is more entangled with (more screened by) electrons at L . Hence $\rho_n(L,T)$ quantifies the entanglement distribution as a function of L at T . We clarify this in the section "Thermal evaporation of cloud shells" of the main text.

Referee #1 comment-7: The formula for ρ_n applied to the charge-Kondo circuit is the same as the formula for \mathcal{N} in Eq. (2) (with M^2 replaced by $\langle \Delta Q/e \rangle^2$). How can \mathcal{N} be replaced by ρ_n (as defined in Eq. (1))?

Response: The excess charge $\Delta Q/e$ plays the role of the magnetization M , since the degenerate charge states of the circuit constitute pseudospin-1/2 states for the Kondo effect. The negativity in the absence of the LSB is $\mathcal{N}_0(T=0) = 1$ at zero temperature. In the presence of the LSB it is $\mathcal{N}(L,T=0;n) = \sqrt{1 - 4M^2/\hbar^2}$. Hence $\rho_n(L,T=0) = \mathcal{N}_0(T=0) - \mathcal{N}(L,T=0;n) = 1 - \sqrt{1 - 4\langle \Delta Q/e \rangle^2} \simeq 2\langle \Delta Q/e \rangle^2$ for small $\langle \Delta Q/e \rangle \ll 1$. At low temperature $T \ll T_K$ (near the fixed point) where thermal fluctuations of $\langle \Delta Q/e \rangle^2$ are negligible, $\rho_n(L,T)$ is approximated as $\rho_n(L,T=0) \simeq 2\langle \Delta Q/e \rangle^2$. We add this explanation in the section "How to detect boundary entanglement shells" of the main text.

Referee #1 comment-8: In the treatment of the channel anisotropic 2CK effects (p14, Supplemental material), the authors use the bosonization approach pioneered by Emery and Kivelson (reference missing!). However, the refermionization in the Emery-Kivelson approach works only for a special value of the z-component of the spin exchange (analogous to the Toulouse limit of the ordinary Kondo problem). I fail to see how the authors have implemented this in their analysis.

Response: We cite Emery and Kivelson (EK) [PRB 46, 10812 (1992)] in the revised manuscript. Regarding the bosonization, we follow the EK approach. We rewrite the total Hamiltonian (including the LSB) in the bosonized form and apply the EK transformation. Then we express the impurity operator in terms of boson fields, following Refs. [S14-S16] where the EK approach is described as change of the boundary condition of boson fields due to the absorption of the impurity. We also express the LSB by boson fields, and compute the impurity magnetization using correlation functions of the boson fields.

As Referee #1 mentions, the EK approach works for a special value of the z-component of the spin exchange. The spin exchange anisotropy does not affect our analysis since the exchange anisotropy is a irrelevant perturbation. By contrast, the LSB is a relevant perturbation that behaves as a magnetic field applied to the impurity. The Kondo temperature T_K and Kondo length ξ_K can be tuned by changing the other spin exchange components so we can take an arbitrary value of L/ξ_K . This analytic approach is confirmed by the NRG calculation.

Referee #2's overall comment: The paper proposes a very intuitive and simple methodology (also novel) to study impurity-bulk entanglement and the cloud structure of Kondo screening. It allows the authors to use both analytical (bosonization) and numerical (NRG) methods to solve the problem, and the approach could be readily applied to other impurity problems. I would recommend publication of this paper but I have remarks which I believe could improve the quality of the paper.

Response: We thank Referee #2 for the positive recommendation and the constructive comments. We revise our manuscript as below, taking into account the comments.

Referee #2 comment-1: the bosonization procedure seems to obtain, in addition to the power-law decay, Friedel oscillations in the entanglement negativity. Such oscillations are absent in the numerics, if I am not mistaken. It would be informative to state why this is the case.

Response: As Referee #2 noticed, the Friedel oscillations were obtained in both of our BCFT and bosonization calculations as shown in Eqs. (S29) and (S34). In Eq. (5) and the figures (numerics) of the main text, we focused on the envelope of the oscillation by setting $2k_F L$ as an integer multiple of π , and found the universality of the Kondo cloud (as a function of L/ξ_K). Note that it would be possible to measure the envelope by fine-tuning $2k_F L$ as in Ref. [21] (Borzenets et al. Nature 2020). To clarify this issue, we add a sentence below Eq. (5) in the main text, sentences below Eq. (S30) and (S34) of Supplementary Information, and another sentence at the end of Supplementary Note 7.

Referee #2 comment-2: The discussion on the shell of anisotropic multichannel Kondo clouds (page 3) is not easy to read and follow. I think the authors should first discuss the different possible scalings : non-Fermi liquid with a certain exponent, Fermi liquid with a Kondo screening and what they call non-Kondo Fermi liquid (it is not straightforward to understand what this last phase is). After defining these different possibilities, a table that shows which scaling applies at which distance -and in which channel - would be more easy to read than the current text.

Response: We thank this comment. Following the comment, we add Table 1 in the section “Isotropic multichannel Kondo clouds” of the main text, to summarize the exponents of the different Kondo models. And we improve the readability of the section “Shells of anisotropic multichannel Kondo clouds” of the main text, and add a paragraph to clarify the non-Kondo Fermi liquid phase in the section.

Referee #2 comment-3: Still on the non-Kondo Fermi liquid, it would be useful to discuss more clearly what this phase/behavior is. In particular, the authors mention an absence of π ($\pi/2$?) phase shift but, as far as I can read, they don't show explicitly this absence.

Response: The presence and absence of the π phase shift have been studied in previous works (e.g., Refs. [45, 46]), which we cite in the main text. We recapitulate the behavior in the channel anisotropic two-channel Kondo model as an example. In the model, a channel more strongly coupled to the impurity shows the phase shift π at $T = 0$ as in the single-channel Kondo effect, while the other weaker-coupling channel does not. This indicates that the weaker-coupling channel does not exhibit the Kondo effect, but showing a normal Fermi liquid. As mentioned in our response to the comment-2 of Referee #2, we add a paragraph to explain this issue.

Referee #3's overall comment: I do not believe that this work contains sufficient new material to be published in Nature Communications -- neither in terms of the methodology nor the physical insights drawn from them.

Response: We thank Referee #3 for her/his efforts of reviewing our manuscript. However we cannot agree with the overall comment. Our work developed (i) a methodology for studying *impurity-bulk entanglement over space* applicable to a wide range of quantum impurities and compatible with various tools (BCFT, bosonization, NRG) and addressed (ii) *quantification of the spatial entanglement distribution* of general multichannel Kondo clouds (physical insights). These have never been studied in previous works including Ref. [41] [Mitchell, Becker, Bulla, PRB (2011)].

The other referees support our work: “addressing the important issue of how to understand the spatial structure of Kondo clouds” (Referee #1); “innovative and careful analysis” (Referee #1); “a new benchmark for theoretical discussions of quantum impurity screening clouds” (Referee #1); “very intuitive and simple methodology (also novel) to study impurity-bulk entanglement and the cloud structure” (Referee #2); “the approach could be readily applied to other impurity problems” (Referee#2).

Moreover, in the arguments of Referee #3, there are points that need clarification (see below).

Referee #3 comment-1: The main message drawn from the study is the 'hierarchical shell' structure of multichannel Kondo models in real-space. However, this is known already from Ref [41], which is cited in the introduction of the present work, but then not mentioned again despite its obvious relevance. Ref [41] does not consider entanglement, as in the present paper, and so there are certainly differences, but the notion of real-space shells around the impurity is not new. To conclude the introduction setting out the main finding: "This shows that different non-Fermi and Fermi liquids hierarchically coexist around the boundary with spatial and energetical separation, reflecting the renormalization of the quantum coherent impurity screening in the presence of the channel competition." However, this is already known. Furthermore, deeper insights are provided in Ref [41]: the shells correspond to real-space regions described by the RG fixed points, with RG flow to lower energies corresponding to real-space flow away from the impurity. In particular, this leads to the conclusion that the shell referred to in this work as the 'core' is actually the local moment shell. This explains the observation in the present work that the core behavior is very similar for multichannel models with different numbers of channels, independent of critical behavior at larger distances ("the bulk does not show any characteristics of the zero-temperature bulk criticality, strongly "binding" with the impurity"). This is because the local moment fixed point is common to all the models considered. Note also that Ref [41] does consider the $k=2$ two-channel Kondo model and discusses the 2CK Kondo cloud, as well as the crossover in real space to Kondo Fermi liquid and non-Kondo shells due to symmetry breaking perturbations.

Response: As aforesaid, the impurity-bulk entanglement [our achievement (i)] was never considered in Ref. [41]. Quantification of the spatial distribution of Kondo clouds by using the entanglement and the LSB [our achievement (ii)] was also not addressed in Ref. [41], as below.

Our impurity-bulk entanglement provides a direct measure of Kondo screening, as it is a key feature of the Kondo singlet states. Hence its spatial distribution quantifies the spatial distribution of Kondo clouds and reveals universal entanglement shells. By contrast, in Ref. [41], Kondo clouds were analyzed by a single-particle quantity, the excess charge density due to the impurity. Figs. 2 and 4 of Ref. [41] showed that the charge density is negative at certain distance from the impurity and that the absolute value of the charge density even increases with the distance. This position dependence cannot be interpreted as the spatial distribution of a Kondo cloud, while it indicates existence of the “local moment” and “strong coupling” spatial regions, providing valuable motivations for further studies.

Our sentences about the hierarchical shells refer the spatial *entanglement* structure in the *tail* of Kondo clouds, as highlighted in the title of our manuscript “hierarchical entanglement shells”. Our work focused on the tail rather than the core, because the entanglement shells in the tail reflects universalities of the BCFTs. We considered three-channel Kondo effects (beyond two-channel cases), to illustrate a tail composed of shells of different non-Fermi liquids. In our revised manuscript, we explicitly mention that the hierarchical shells refer a spatial *entanglement* structure, and cite Ref. [41] in relevant locations.

Referee #3 comment-2: Ref [41] also considered the "thermal evaporation" of the shells at finite-T.

Response: This comment needs clarification. Ref. [41] considered thermal evaporation only for the single-channel Anderson impurity model; it never studied thermal evaporation of multichannel Kondo effects (a topic of our manuscript) including two-channel cases. We explicitly mention this in the section “Thermal evaporation of cloud shells” of the main text.

Referee #3 comment-3: In terms of methodology, again the present work does not really do something very new. The NRG technique used is sophisticated, but was developed in Ref [30] already, and used several times since to apprehend entanglement negativity properties of Kondo problems. Eq. 2 relates the $T=0$ negativity to magnetization, which is a very nice result -- but this was derived already in Ref [31]. In Ref [31], finite-T negativity between the impurity and the channels for multi-channel Kondo models was already studied. The innovation here is to study this same quantity with the same methods, but now with a local perturbation at a distance L away from the impurity. This gives the possibility to discuss negativity changes as a function of the length scale L . Something similar was proposed in Refs [19,20] to probe the Kondo cloud (there the focus was measurable conductance, here the authors discuss the negativity).

Response: We cannot agree with this comment. Our work developed a novel approach for studying the spatial distribution of impurity-bulk entanglement by introducing and handling a *local* $SU(2)$ *symmetry breaking* perturbation (LSB). This new methodology is applicable to boundary-bulk entanglement in general spin-1/2 BCFTs (boundary conformal field theory) and other impurity problems. This achievement is strongly supported by the other referees, as aforesaid.

None of the papers (including Ref. [41]) mentioned by Referee #3 considered the LSB. The LSB should be distinguished from perturbations (those in Refs. [19,20]) not breaking the $SU(2)$ symmetry. The latter does not cause any reduction of Kondo screening at zero temperature, while the LSB does; hence the latter is not useful for studying the spatial distribution of impurity-bulk entanglement.

Referee #3 comment-4: Finally, the authors propose and experiment to measure the entanglement signatures discussed in the paper. However, this potentially interesting section is not elaborated on much and is hard to follow, especially for non-experts. The main message however, is that a **charge-Kondo** setup allows to get an estimate on the negativity via dot charge changes, with and without a perturbation in the leads. This is an interesting idea, but unfortunately I do not think is practical, because the charge differences in the regime of interest are extremely small (that is because the interesting power-laws in ρ_n set in at large L , which means very small ρ_n in practice). This would be well below the charge detection noise level in an actual experiment. Furthermore, the scaling requires changing L , but a real device would have a fixed quantum point contact at a specific L , which is a physical property of the constructed device. One could make several different devices with different L , but the other inevitable differences between different devices would confound any comparison -- again especially when considering the very small numbers involved. Therefore, the statements by the authors that "suppression of the entanglement can be ... experimentally detected" and "The boundary-bulk entanglement will be

experimentally accessible" are wildly overstated. This would in practice be very difficult or even impossible, and there is certainly no evidence that this is doable.

Response: We expect that our proposal based on the charge-Kondo setup is within experimental reach. In Supplementary Note 7, we already discussed estimation of parameters of the setup, the experimental feasibility, and how to access the scaling behavior (the dependence on L) with a single device. Regarding its feasibility and detecting sensitivity, we have discussed with Frederic Pierre, a leading expert on the experiments (see the footnote [S20] "F. Pierre, private communication"). He confirmed us that our proposal is within experimental reach with desired accuracy of excess charge measurement. We revise this section of the main text to be more understandable for non-experts.

About the other issue on observing the scaling behavior with changing the location L of the LSB, a recent work demonstrated variation of the distance L with respect to the Kondo cloud length, using a *single* device [see Fig. 3 of Ref. [20], Borzenets et al., Nature 579, 210 (2020)]. This approach is applicable to our proposed setup, as we already discussed in Supplementary Note 7.

Referee #3 comment-5: With $\rho_n(L,T)$ the negativity difference with and without the perturbation at site L , the authors state: "Larger ρ_n implies that at the distance L there exist more electrons participating in the entanglement." This seems crucial to the conclusions drawn, but is unsubstantiated. The negativity calculated is that of the impurity with the entire baths; adding the perturbation at position L can change the entanglement profile of the channels in real-space, which is then traced out. It is not clear to me that a perturbation at position L tells us about entanglement at length scale L . Philosophically, we can incorporate the perturbation at position L into the definition of a new non-interacting conduction electron bath, which will then have an energy-dependent hybridization function coupling to the impurity. The perturbation at L will lead (in 1d, which I guess is the setup used) to a feature at an energy scale $\omega_p \sim 1/L$ in the hybridization function. If this perturbation scale ω_p is much smaller than the Kondo scale T_K , then there will be very little effect. If $\omega_p \gg T_K$ then the Kondo effect will be affected. It seems that one can argue the results based only the energy-structure of the hybridization function, and don't need to invoke a length-scale at all! For example, one would get the same ρ_n and deduced entanglement structure in a different system that just happens to have the same energy-dependent hybridization function, where the length scale L has no meaning or analogue.

Response: This comment needs clarification. Nothing is traced out in the evaluation of the negativity. And, Referee #3 did not mention about the LSB, the essential ingredient of our approach for studying impurity-bulk entanglement, in this comment and the comment-3.

The LSB breaks the SU(2) Kondo spin symmetry at distance L . It is obvious that the LSB reduces the Kondo screening (if exists) by local electrons at the distance. The quantity ρ_n is the difference of the impurity-bulk entanglement between the absence and presence of the LSB. Larger value of ρ_n is directly interpreted as more electrons participating in the Kondo screening cloud. Hence the dependence of ρ_n on the distance L quantifies the spatial distribution of the cloud. The simple conversion to energy ($\sim 1/L$) is not enough for explaining the local suppression of Kondo screening by the LSB (symmetry breaking), differently from perturbations not breaking the SU(2) symmetry.

Referee #3 comment-6-1) the language and presentation needs to be improved throughout

Response: We thank this comment. We revise our manuscript to be more readable and understandable.

Referee #3 comment-6-2) In the intro: "While bulk quantities have been understood, boundary states are yet to be explored." This is not true, they have been studied in the existing literature, as mentioned above, and in other works.

Response: The sentence is written for general quantum impurity problems and BCFTs. While the bulk quantities have been understood much, the boundary states (e.g., its key feature such as the boundary-bulk entanglement) remain largely unexplored.

Referee #3 comment-6-3) In the intro: "The properties of the cloud, such as its channel-resolved spatial distribution ... are yet to be studied." This is wrong, it has been studied before, with many of the same conclusions! See e.g. [41].

Response: We believe that the sentence was properly and carefully written. We refer the full sentence, restoring the ellipsis part: "The properties of the cloud, such as its channel-resolved spatial distribution, its entanglement with the impurity, its correspondence to the transition or crossover between distinct non-Fermi liquid phases, and its thermal suppression, are yet to be studied". Ref. [41] did never discuss about quantification of the spatial cloud distribution, entanglement, crossover between distinct non-Fermi liquid phases (which appears in three-channel Kondo effects), and thermal suppression of multichannel Kondo clouds (see also our responses to Referee #3 comments-1&2).

Referee #3 comment-6-4) The form of the conduction electron Hamiltonians should be specified. In particular, are the authors explicitly considering 1d systems? This should be mentioned if so!

Response: We consider 1D systems, following the usual mapping of Kondo problems onto 1D. This is now explicitly mentioned in the revised version of our manuscript.

Referee #3 comment-6-5) The LSB consists of a field B in the bath. What is the strength of B , and how do the results depend on B ? If B is supposed to be infinitesimal, how is this dealt with in NRG? This should be mentioned in the main text.

Response: Supplementary Information already contains these technical points. For example, the strength of B is set $0.1D$ with the half band width D (the last paragraphs of Supplementary Note 2).

Referee #3 comment-6-6) "In the core, ρ_n rapidly decays with L , showing that most electrons forming the cloud are in the core." I am not convinced this is implied. And furthermore, the decay of ρ_n is more rapid in the tails, as shown in Fig 1! The power law involves a more negative power in the tail than core, so surely its decay is more rapid *outside*. I would say the region inside the core is the "local moment cloud".

Response: We thank the comment. To avoid possible misunderstanding, we revise the sentence: " ρ_n is much larger in the core than in the tail, showing that most electrons forming the cloud lies in the core".

Referee #3 comment-6-7) Original references should be given to published literature where the crossovers in 3CK to either 1CK or 2CK depending on the sign of δJ are discussed. This is not a new finding.

Response: We thank Referee #3 for this comment. We cite the previous work in Ref. [37] on the sentence.

Referee #3 comment-6-8) In the conclusion, it is stated that "In spin-1/2 boundary criticalities, [the entanglement structure] is obtained using the boundary magnetization, Eq 2." This is only true at $T=0$ and not general. The finite- T result cannot be simply related to a boundary observable.

Response: We already mentioned that Eq. (2) (Eq. (3) after the revision) is valid at $T = 0$ and a good approximation at low T (see the paragraph below Eq. (3) of p2).

REVIEWERS' COMMENTS

Reviewer #1 (Remarks to the Author):

I have studied the revised version of ms NCOMMS-22-39861A, as well as the authors' response to the comments and questions in my original report. In my opinion, all issues that I brought up in this report have been addressed satisfactorily.

I recommend publication in Nat. Commun.

Reviewer #2 (Remarks to the Author):

The authors have convincingly addressed all my comments and queries. Their responses to referees 1 and 3 were also well-argued and convincing to me.

I thus believe that the work should be published in Nature Communications.

Reviewer #3 (Remarks to the Author):

The major concern is that the hierarchical shell structure of the single and multichannel Kondo models is already quite well known. Furthermore, since this shell structure results from the universal properties of the renormalization group (RG) flow in real space, it is entirely expected that all physical properties and observables exhibit this same shell structure. The real-space shell structure associated with the different RG fixed points has been studied in the single and channel-symmetric/asymmetric multichannel cases, and the 'evaporation' of the Kondo cloud at finite temperature has also been studied.

It is true that the *entanglement* properties associated with the hierarchical shell structure have not been studied before using this method before. Note that I mention this in my original review: "Ref [41] does not consider entanglement, as in the present paper, and so there are certainly

differences, but the notion of real-space shells around the impurity is not new." The specific entanglement measure considered in this work and applied to these specific problems is new, but the conclusion about the shell structure is already known and that this is reflected in entanglement is completely expected. What new insights does this study bring? My worry is that new insights are not really gained, but the text of the article seems to imply that the authors have discovered a shell structure. The flow of entanglement from one regime to another in real space, in multichannel Kondo models, at finite temperature, in real-space, has been considered before -- see e.g. PRL 114, 057203 (2015) by one of the same authors as the present work. Here we just have a different entanglement measure. In terms of techniques, many of the same tools have been developed and used before. The fact that local symmetry-breaking is introduced here is nice, but hardly a crucial breakthrough. I am not saying this is of no value, just that the originality and importance of the result are overstated. The work is more suited to a more specialized journal

Reply to Referee #1

We would like to thank Referee #1 for reviewing our manuscript and recommending its publication.

Reply to Referee #2

We very much appreciate the review of our manuscript and the positive recommendation by Referee #2.

Reply to Referee #3

Comment of Referee #3 in her/his second report: “The major concern is that the hierarchical shell structure of the single and multichannel Kondo models is already quite well known. Furthermore, since this shell structure results from the universal properties of the renormalization group (RG) flow in real space, it is entirely expected that all physical properties and observables exhibit this same shell structure. The real-space shell structure associated with the different RG fixed points has been studied in the single and channel-symmetric/asymmetric multichannel cases, and the 'evaporation' of the Kondo cloud at finite temperature has also been studied.

It is true that the *entanglement* properties associated with the hierarchical shell structure have not been studied before using this method before. Note that I mention this in my original review: "Ref [41] does not consider entanglement, as in the present paper, and so there are certainly differences, but the notion of real-space shells around the impurity is not new." The specific entanglement measure considered in this work and applied to these specific problems is new, but the conclusion about the shell structure is already known and that this is reflected in entanglement is completely expected. What new insights does this study bring? My worry is that new insights are not really gained, but the text of the article seems to imply that the authors have discovered a shell structure. The flow of entanglement from one regime to another in real space, in multichannel Kondo models, at finite temperature, in real-space, has been considered before -- see e.g. PRL 114, 057203 (2015) by one of the same authors as the present work. Here we just have a different entanglement measure. In terms of techniques, many of the same tools have been developed and used before. The fact that local symmetry-breaking is introduced here is nice, but hardly a crucial breakthrough. I am not saying this is of no value, just that the originality and importance of the result are overstated. The work is more suited to a more specialized journal”

Response: Referee #3 repeated essentially the same concerns with her/his first report. We have already responded to those concerns in our previous response. The other referees supported our arguments, and continued to recommend publication of our manuscript in Nature Communications. Especially, Referee #2 mentioned “Their responses to referees 1 and 3 were also well-argued and convincing to me”.

We must point out that there are incorrect statements by Referee #3, “The flow of entanglement from one regime to another in real space, in multichannel Kondo models, at finite temperature, in real-space, has been considered before -- see e.g. PRL 114, 057203 (2015) by one of the same authors as the present work. Here we just have a different entanglement measure. In terms of techniques, many of the same tools have been developed and used before.”. Contrary to the statements, the real-space structure of multichannel Kondo models (the central theme of the present manuscript) was never considered in the previous work [PRL 114, 057203 (2015)]. Moreover, the conformal field theory (the tool for obtaining analytic expressions in the present manuscript) was never used in the previous work.

We slightly revise our manuscript, to acknowledge Ref. [41] and clarify the novelties of our work.